# Culturing the Human Oral Microbiota, Updating Methodologies and Cultivation Techniques

**DOI:** 10.3390/microorganisms11040836

**Published:** 2023-03-24

**Authors:** Saber Khelaifia, Pilliol Virginie, Souad Belkacemi, Herve Tassery, Elodie Terrer, Gérard Aboudharam

**Affiliations:** 1Institut Hospitalo-Universitaire Méditerranée-Infection, Aix-Marseille Univ, IRD, MEPHI, AP-HM, 19–21 Boulevard Jean Moulin, 13385 Marseille CEDEX 05, France; 2Ecole de Médecine Dentaire, 27 Boulevard Jean Moulin, 13385 Marseille CEDEX 05, France

**Keywords:** culture, human oral microbiota, bacteria, archaea, fungi, culturomics

## Abstract

Recent years have been marked by a paradigm shift in the study of the human microbiota, with a re-emergence of culture-dependent approaches. Numerous studies have been devoted to the human microbiota, while studies on the oral microbiota still remain limited. Indeed, various techniques described in the literature may enable an exhaustive study of the microbial composition of a complex ecosystem. In this article, we report different methodologies and culture media described in the literature that can be applied to study the oral microbiota by culture. We report on specific methodologies for targeted culture and specific culture techniques and selection methodologies for cultivating members of the three kingdoms of life commonly found in the human oral cavity, namely, eukaryota, bacteria and archaea. This bibliographic review aims to bring together the various techniques described in the literature, enabling a comprehensive study of the oral microbiota in order to demonstrate its involvement in oral health and diseases.

## 1. Introduction

The human oral cavity is a complex ecosystem favourable to the development and establishment of numerous bacterial, archaeal, viral, fungal and unicellular eukaryota species [1]. These different kingdoms of life cohabit and evolve within this complex ecosystem, composing what is commonly known as “microbiota” [2]. This is represented by a complex community that persists and develops on the surfaces of the oral cavity, generally in the form of a multi-species biofilm which transits through the saliva [3]. This biofilm, when growing on the teeth, is referred to by the generic term “dental plaque” [3]. The unique properties of the oral cavity and its constituent organs make the composition of the oral microbiota characteristic of the site it colonises and distinguishes it from that of neighbouring ecosystems, such as the skin, respiratory and digestive tracts [4]. These properties play a major role in determining the oral microbiota and its composition [4]. Several parameters are involved in determining which species can colonise, grow and settle to become a major or minor component of the microbiota, including the quality and quantity of saliva, the biotype and phenotype of the periodontium, the depth of the periodontal pockets around the teeth and the quality of the dental tissue [5].

The oral microbiota maintains a symbiotic relationship with its host [6]. The resident microorganisms of the oral cavity help fight pathogens, down-regulate undesirable and potentially pro-inflammatory responses, and promote oral health [7,8]. However, this symbiotic relationship remains fragile and susceptible to change depending on social, economic, dietary and nutritional factors; sedentary lifestyle, autonomy, hygiene compliance, smoking and alcohol consumption as well as genetic and physiological factors [5,9,10,11,12]. These factors directly influence the composition of the microbiota and also its behaviour and its evolution from a commensal microbiota to pathogenicity [13]. Since the notion of infectious disease first appeared, humans have tried to classify microorganisms as pathogens or commensals as part of the local and resident microbiota, evolving in perfect symbiosis with their host [14]. It is now known that some bacterial species or even communities may have properties that escape our classification systems [15]. The distinction between pathogenic and commensal bacteria is becoming an increasingly complex and ambiguous task, especially when the microorganisms usually found in healthy states are involved in complex interactions with their host leading to a state of disease or infection [15]. Recent years have been marked by a paradigm shift in the study of the human microbiota, with a re-emergence of culture-dependent approaches [16,17,18,19,20,21]. Recent studies have been mainly devoted to cultivating the gut microbiota and studies on the oral microbiota remain very limited. The oral microbiome is highly diverse. Approximately 774 indigenous species are present in the oral cavity, including 54% cultivated and named species, 14% cultivated but not yet named species, and 32% remaining uncultivated phylotypes [6] (the Human Oral Microbiome Database, https://www.homd.org/; accessed on 23 January 2023). Recently, molecular-based studies have demonstrated that the oral microbiota is more diverse and complex and plays a vital role in the pathogenesis and development of many oral and systemic diseases [22,23].

In this literature review we review new strategies that have been developed to improve the culture and isolation of microbes from the human microbiota that can be applied to cultivate the oral microbiota, as well as their field of application.

## 2. Current Strategies to Study the Human Oral Microbiota

### 2.1. Next-Generation Sequencing and Metagenomics

Next-generation sequencing (NGS) is a new molecular approach applied to study the microbial diversity of diverse ecological niches [24,25]. NGS has enabled a more in-depth and non-specific exploration of the human microbiota by generating a huge amount of data leading to a better understanding its composition, mainly through amplicon sequencing and whole-genome metagenomics [26,27]. NGS has enabled a rapid and comprehensive analysis of microbial diversity in various ecological niches, which has propelled it to the forefront to gradually replace the tedious and time-consuming standard culture techniques [24,28].

However, the limitations of NGS are quickly identified. Indeed, the absence of a standardised protocol and the variability of the used DNA extraction techniques, as well as the variability of the primers used for amplification, create biases that make the results dependent on the technique used [25]. In addition, the depth bias associated with all molecular methods, due to limited sensitivity, results in the neglect of minority species [29]. One of the major shortcomings of molecular techniques remains the lack of information related to the viability, adaptation, and availability of the strains for further in vivo studies [18,21].

The limitations and lack of information generated by molecular techniques encourage us to return to the sources of the fundamental research in microbiology, namely microbial culture [18]. Indeed, microbial culture represents the basic component of microbiology, and has long been side-lined at the expense of molecular methods [18,24]. Much recent work underlines the need to combine molecular techniques such as NGS with recent and innovative approaches such as culturomics in order to overcome the lack of these techniques and to gain a global and exhaustive view of the composition of the microbiota of a given ecological niche [21].

### 2.2. Culturomics: A New Concept Applicable to Oral Microbiota

Culturomics is a high-throughput culture-based concept created in 2012 with the goal of isolating all species that make up the human digestive microbiota [30]. Since then, many studies have combined high-throughput culture and metagenomics with the goal of reducing the dark matter of the microbiota, consisting of unassigned and unmatched sequences, through the isolation and sequencing of previously uncultured species [20].

Culturomics has proven itself over the last ten years, and its application has been made possible by rapid colony identification using Matrix Assisted Laser Desorption Ionization/Time Of Flight Mass Spectrometry (MALDI-TOF MS) [30]. Culturomics has demonstrated its effectiveness and contribution to the exploration of the gut microbiota over the last decade by significantly expanding the field of knowledge regarding the repertoire of the human microbiota by isolating new or previously uncultured species [18,19,20,21,30]. Indeed, to date, more than 2770 different species, including 800 new species, have been isolated by culturomics from human specimens from different sites including: human gut, skin, urinary tract, vaginal and respiratory microbiota [31].

Various studies throughout the world have shown the interest of diversifying culture media and conditions to broaden the scope of culture, in addition to the use of specific media and targeted approaches (Table 1). Indeed, the use of selective and non-selective media has allowed a continuous progression of knowledge concerning the diversity of the human microbiota [32]. The oral microbiota is undoubtedly very diverse, as shown by the Human Oral Microbiome Data Base (https://www.homd.org/; accessed on 23 January 2023), which currently contains more than 774 different species (https://www.homd.org/; accessed on 23 January 2023) (Figure 1) [33,34]. Currently, metagenomics-based estimates suggest a global number of around 1200 species grouped into 19 different phyla, namely *Absconditabacteria* (SR1), *Actinobacteria*, *Bacteroidetes*, *Chlamydiae*, *Chlorobi*, *Chloroflexi*, *Cyanobacteria*, *Firmicutes*, *Fusobacteria*, *Gracillibacteria* _(NG02), *Ignavibacteriae*, *Lentisphaerae*, *Proteobacteria*, *Saccharibacteria*, *Spirochaetes*, *Synergistetes*, *Tenericutes* WPS-2, and *Euryarchaeota* (https://www.homd.org/taxa/tax_hierarchy; accessed on 23 January 2023) [22]. The majority of cultured species are grouped in only six phyla, *Actinobacteria*, *Bacteroidetes*, *Firmicutes*, *Fusobacteria*, *Proteobacteria*, and *Spirochaetes* (Figure 2) [22].

The oral cavity is the main entrance and an obligatory passage for colonising microbes ending their journey in the digestive tract. The colonisation of these microbes to the oral ecosystem is also possible thanks to the favourable living conditions for the development and establishment of various microorganisms including bacteria, archaea, fungi, viruses, giant viruses, Candidate Phyla radiation group (CPR) and unicellular eukaryotes. Microbial culturomics may be a helpful tool for studying the oral microbiota by culture in order to broaden the spectrum of cultivated species of this ecosystem.

### 2.3. Rapid Culture of Anaerobic Bacteria Using YCFA Medium

Yeast extract-casein hydrolysate fatty acids (YCFA) culture medium is a rich medium widely used in the literature to culture strict anaerobes [35,36]. It is composed of growth factors, antioxidants, volatile fatty acids and vitamins essential for their growth (https://www.dsmz.de/microorganisms/medium/pdf/DSMZ_Medium1611.pdf; accessed on 23 January 2023).

One very interesting approach to cultivating anaerobic bacteria in record time was proposed by Naud et al., in 2020, using YCFA medium as well as the medium of blood culture bottles (BioMérieux, Marcy l’Etoile, France) supplemented with rumen [36]. The culture investigations, which were performed on only two stool samples, resulted in the isolation of 121 different bacterial species in only three weeks, of which 104 species were cultured within 24 h of incubation (Table 1). This is an unprecedented technical achievement, making it possible to use bacterial culture as a rapid and accurate diagnostic tool [36].

According to another study conducted by Browne et al., the YCFA culture medium allowed the cultivation of 137 species, 68 of which were completely new species, and 63 of which belonged to the phylum *Firmicutes*, four to *Bacteroidetes* and one to *Actinobacteria* [35]. This medium also gave exceptional results during a study by Forster et al. in 2019 on the diversity of the human digestive macrobiota, by succeeding in cultivating 273 different bacterial species, of which 105 were new bacterial species distributed over three phyla: 91 species belonged to the phylum *Firmicutes*, 13 to the phylum *Bacteroidetes*, and one to the phylum *Proteobacteria* [37].

The oral microbiota consists of a large majority of anaerobic microorganisms [38,39]. YCFA medium is suitable for the culture of these oxygen-intolerant microorganisms and the studies described above demonstrate the high yield of culture using this medium to cultivate anaerobes [35,36]. Moreover, this medium is able to cultivate the most represented phyla in the human oral cavity, namely *Firmicutes*, *Bacteroidetes*, *Actinobacteria* and *Proteobacteria* [35,36]. The YCFA medium may constitute a good culture base for future investigations to explore the diversity of the oral microbiota. So far, and according to the sources we have consulted, no study exploiting the potential of the YCFA medium to study the diversity of the oral microbiota has been conducted.

### 2.4. Specific Conditions and Media for a Targeted Culture

#### 2.4.1. Spore-Forming Bacteria in the Oral Cavity

Recent studies have described the isolation and characterisation of spore-forming bacteria from the oral cavity in the context of pathogenicity [40] or as an integral part of the oral microbiota [41]. Despite these limited studies, the number of spore-forming bacteria colonising the oral cavity is limited to a few cultivated species, mainly from the genera Bacillus or *Peanibacillus* [41]. Numerous techniques have been optimised for the specific culture of spore-forming bacteria in various domains, including the intestinal microbiota (Table 1).

#### 2.4.2. Alcohol Decontamination

In 2016, Browne et al. described the use of improved pre-culture conditions that allowed for the targeted isolation of spore-forming bacteria [35]. This technique consisted of pre-incubating the sample in an equal volume of 70% alcohol for four hours at room temperature under aerobic conditions (Table 1). A wash was then performed prior to plating on a rich culture medium under anaerobic conditions [35].

The alcohol selection or disinfection technique was then adapted and modified by the culturomics team in 2020 to specifically isolate spore-forming bacteria that could be used in bacteriotherapy trials for the treatment of *Clostridium difficile* infections from human stool samples [42]. A total of 254 bacterial species were identified, of which nine represented new species. Of the species isolated in this study, 242 have never been included in clinical trials and represent potentially interesting new candidates for bacterial therapy trials [42].

#### 2.4.3. Thermal Shock

Thermal shock is a selective technique based on the use of high temperature for the isolation of spore-forming bacteria. One of the culture protocols validated by the culturomics approach is to expose the sample to a constant temperature of 80 °C in a dry bath for 20 min to select spore-forming or spore-producing bacteria. The exposed sample is then subcultured on Columbia agar plates in anaerobic conditions [18].

#### 2.4.4. The Use of Antioxidants to Improve Anaerobic Conditions

Antioxidants have long been used to establish an anaerobic atmosphere or to contribute to its improvement [43,44]. Indeed, antioxidants such as sodium sulphide (Na_2_S), or amino acids such as cysteine, are found in old culture media optimised for the culture of anaerobic bacteria but also for the culture of methanogenic archaea for which sensitivity to oxygen is widely demonstrated [17,43,44]. Since then, many studies have focused on the beneficial effect of antioxidants for the culture of fastidious anaerobes. A more recent study conducted by La Scola et al., in 2014, succeeded in cultivating six strictly anaerobic bacteria and seven aerobic bacteria aerobically on Schaedler agar culture medium supplemented with 0.1 g glutathione and 0.5 g ascorbic acid. Among the species cultivated during this study, the authors mention the *Fusobacterium necrophorum* and *Ruminococcus gravus* species, known for their extreme sensitivity to oxygen [45]. Their results suggest that using this unique medium it would be possible to cultivate anaerobic and aerobic bacteria aerobically in a standard incubator, enabling the renewal of the anaerobic culture [45].

In 2015, Dione et al. were able to expand the spectrum of bacterial species grown with antioxidants by culturing aerobic, microaerophilic, and anaerobic bacteria on a single medium by a simple addition of antioxidant consisting of 0.1 g glutathione +1 g uric acid +1 g ascorbic acid to Shedler agar medium [46]. This technical feat enabled the culture of 251 different bacterial species and showed that it was possible to perform antibiograms of anaerobic bacteria in aerobic conditions without affecting the sensitivity of the results [46]. This technique was then applied to methanogenic archaea culture to successfully isolate 13 strains of *Methanobrevibacter smithii* and nine strains of *Methanobrevibacter oralis* from 100 stools and 45 oral samples [16].

The work of Million et al., in 2020 confirmed the beneficial effect of antioxidants on the culture of anaerobes [47]. In this case, antioxidants were able to decrease the oxidative stress experienced by anaerobic bacteria of the genus *Clostridium* and maintain an increased production of volatile fatty acids such as butyrate, isobutyrate and isovalerate when grown in the presence of three antioxidant molecules: glutathione, ascorbic acid and uric acid. Volatile fatty acids are known to contribute to the maintenance and active resilience of host–bacterial mutualism against mucosal oxygen and uncontrolled oxidative stress in vivo [47].

Recently, new molecules with antioxidant activities have been described in the literature, such as phenolics, flavonoids and carotenoids [48]. These molecules with antioxidant potential should be tested for their antioxidant effect in order to improve or even establish the anaerobic atmosphere necessary for the culture of anaerobic bacteria in the oral cavity.

#### 2.4.5. Bacterial Co-Culture

Co-culture has been widely used in recent years to make possible the culture of certain microorganisms that so far had remained uncultured. Co-culture is based on the capacity of some bacteria to produce or secrete active substances, cofactors or growth factors of interest to other species lacking them [16]. Numerous approaches have been used either to accelerate the bacterial growth, by the addition of missing unknown nutrients in the culture medium, or of known nutrients not commercially available [16,49,50,51]. To illustrate the contribution of co-culture to the field of microbial culture, and in particular oral cavity microbes, we took the example of methanogenic archaea.

The culture of methanogenic archaea is fastidious and requires an external supply of H_2_ and CO_2_ required for their growth [17]. In 2016, Khelaifia et al. succeeded in substituting the gaseous supply of H_2_ and CO_2_ by a co-culture with *Bacteroides thetaiotaomicron* used as the sole source of hydrogen and CO_2_ in a dual chamber culture device. In this device, *B. thetaiotaomicron* were grown in the bottom chamber in Schaedler liquid medium supplemented by antioxidants producing enough hydrogen and CO_2_ to allow growth of the methanogenic archaea *Mathanobrevibacter smithii* on solid medium and isolate viable colonies [16]. Previous studies have used cultured species to boost the growth of other fastidious bacteria using diffusion chambers or a double layer of agar separated by porous membranes [49,50]. Co-culture is therefore becoming a good alternative for laboratories lacking the technical means to culture fastidious bacteria and methanogenic archaea from all sources, whether biological or environmental [16].

#### 2.4.6. Selective Media Based on Antimicrobial Agents, Antibiotics and Phages

The ability of some microorganisms, including bacteria and archaea, to resist antimicrobial agents is one of the most important public health problems of recent years, but also poses an interesting possibility for microbiologists to decontaminate samples for the purpose of the specific isolation of these resistant microorganisms [52,53]. Methanogenic archaea are known to be resistant to the majority of antibiotics used in antimicrobial therapeutics and the treatment of infectious diseases, except for metronidazole derivatives, chloramphenicol and fusidic acid [54,55]. Many studies have focused on this ability found in some microorganisms to facilitate their isolation from a complex environment. The culture medium optimised by Khelaifia et al., in 2013, is now one of the most widely used media for the selective culture of methanogenic archaea [16] (Table 1). One of the peculiarities of this medium, in addition to its complex nutrient composition (thus considered a rich medium), is the addition of a cocktail of antibiotics aimed at eliminating Gram-positive and Gram-negative bacteria, fungi and yeasts [16,54].

The disadvantage of the use of antibiotics is the obligatory selection of resistant strains [52,53]. Bacterial growth can also be inhibited by the use of phages, which reduces the potential development of resistance associated with the use of antibiotics [56]. Indeed, bacteriophages are viruses that specifically infect certain bacteria. In the case of lytic phages, host bacteria are destroyed by their phage through a process of cell lysis [18]. This specific lysis capacity can therefore be used to specifically eliminate majority bacteria in a polymicrobial culture of any origin, including the human oral cavity, in order to cultivate minority and slow growing bacteria.

#### 2.4.7. The Genomic-Reverse Culture Technique

Reverse genomics is one of the most promising new perspectives in the cultivation of uncultured microorganisms in all areas of life [57]. This technique exploits the ability of antibodies to bind specifically to a given organism, to isolate it by culture, and then sequence its genome from a simple culture or complex ecosystem [57]. Using this method, Cross et al. successfully isolated and cultured three species of *Saccharibacteria*, including the TM7 strain known to be an integral part of the human oral macrobiota [57]. They also isolated and cultured species from the human oral SR1 bacteria group, which are members of a previously uncultured bacterial lineage [57].

A more recent study has also demonstrated the potential of reverse genomics by identifying a common system capable of culturing all members of the *Saccharibacteria* group from any human sample [58]. This method is also applicable to any other sample or specimen of diverse origin [58]. In this study, the authors describe their success in obtaining in-silico specific epitopes for *Saccharibacteria* spp., distributed in four different transmembrane proteins. Subsequently, they suggest the use of all these epitopes to synthesise antibodies targeting the Saccharibacteria species in order to cultivate them. According to the authors, this strategy can also be applied to archaea, or to other phyla/taxa, such as the Parcubacteria phylum and the DPANN group (*Diapherotrites*, *Parvarchaeota*, *Aenigmarchaeota*, *Nanoarchaeota*, and *Nanohaloarchaeota*) [58]. Thus, the use of this methodology can help cultivate new microbes and enrich our knowledge of the composition of the human microbiota in general and the oral microbiota in particular. Most microorganisms in the human oral cavity from all taxonomic levels remain uncultured to date [59]. Reverse genomics can be a significant asset for the cultivation of uncultured microorganisms and can be applied to all species from all known ecosystems [57]. These species, once isolated and cultured, can be characterised and studied to better understand both the role they play in the oral microbiota but also their actual involvement in oral disease and health [60].

### 2.5. Treponemes of the Oral Cavity

Treponemes are spiral-shaped motile prokaryotes belonging to the genus *Treponema* and the order Spirochaetales, in the family *Spirochaetaceae* [61,62]. These microorganisms colonise the human intestinal [63], oral [64] and genital [65] microbiota, as well as those of animals. Most of these microorganisms are part of the commensal microbiota, but some species are opportunistic pathogens, such as *Treponema pallidum*, the causative agent of syphilis in humans [66]. Other species have been directly associated with a high prevalence in patients with periodontitis and gingivitis, such as *Treponema denticola*. Indeed, numerous studies have revealed the direct involvement of certain *Treponema* species in the severity and extent of periodontal lesions [67,68]. *T. denticola* is the most widely studied of these, and its virulence factors have been identified. This bacterium is a member of the Sokransky red complex [69] and has the ability to bind to fibroblasts and epithelial cells via dentilisin protease, thus becoming cytotoxic and causing cell death [70].

Treponemes are a typical example of fastidious microorganisms with complex nutritional requirements. They cannot grow on minimal media, and their culture requires special and complex nutrients. Most *Treponema* species grow at temperatures ranging from 37 °C to 40 °C, anaerobically and/or micro-aerobically [71]. Several culture media have been reported in the literature, specifically designed to support the growth of these bacteria, and several modifications have been made to one or more of these media, including the addition of vitamins, carbohydrates, volatile fatty acids and serum of animal or human origin (Table 1) [71]. Initially, media and techniques previously designed for the culture of *T. pallidum* led to the isolation of the first treponemes associated with the human genital mucosa (*Treponema calligyrum*, *Treponema refringens*) [72] and the oral cavity (*T. denticola*) [73]. Thus, the nutritional requirements deduced from attempts to grow *T. pallidum* led to the formulation of specific culture media, adapted to the culture of *Treponema* spp., such as Growth Medium (GM-1) [74], Enriched Medium 10 (M10) [75], Oral Treponemes Isolation medium (OTI) [76], New oral spirochete medium (NOS) [77], Oral Treponemes Enrichment Broth (OTEB) [78], Oral Microbiology and Immunology, Zürich medium (OMIZ) [79], and more recently T-Raoult [71]. Teponemes culture media are summarized in Table 1.

The genus *Treponema* is the predominant genus of *Spirochaetaceae* colonising the oral cavity. Although these organisms are part of the normal microbiota, they have, to date, been neglected by microbiologists, and the number of cultured species is thus far limited to only ten species [71]. However, studies based on molecular methods by direct amplification of 16S DNA from oral cavity samples have revealed the existence of more than 80 different phylotypes belonging to the genus *Treponema* that remain uncultured so far [80]. Hence, there is a pressing need to develop new identification and culture methods specifically targeting *Treponema* from clinical samples, to allow the extension of the *Treponema* repertoire associated with human microbiota and, more particularly, the oral microbiota. This can be achieved by isolating new species to phenotypically characterise them and sequence their genomes in order to study their roles in human oral health and pathology to determine their role within this complex microbiota.

### 2.6. Archaea of the Oral Cavity

Archaea are unicellular prokaryotes forming a separate domain of life from bacteria, eukaryotes and giant viruses [81]. They were first isolated from extreme environments, such as hot springs [82] and salty lakes, but it is now known that they are ubiquitous and can be found in all known ecosystems, including living organisms [83,84]. In humans, methanogenic and some halophilic archaea species are principally found [85,86]. Archaea have colonised all human mucosa, particularly intestinal [87] and oral mucosa [88], and have been associated with dysbiosis [89] as well as abscessed pathologies [90].

Culturing methanogenic archaea is a long and fastidious process, due to their oxygen intolerance and specific nutritional needs [17]. Most methanogenic archaea require an external source of hydrogen and carbon dioxide for their growth [17]. In 1969, Hungate pioneered methanogenic archaea culture. He first did so in a strict anaerobic liquid medium, and then in a solid medium in roll tubes, which allowed a control of the culture atmosphere in general, as well as the addition of hydrogen and carbon dioxide at various concentrations in particular [44]. Since then, different media appropriate to the requirements of each known species have been optimised (Table 1) [17]. Recently, the SAB^®^ medium has been developed and optimised in order to be versatile and enable the culture and isolation of most methanogenic archaea species [17]. The addition of antioxidants to agar SAB^®^ medium also enabled the aerobic culture of methanogenic archaea, by placing the agar plates in the upper part of a double chamber, with a lower part containing liquid culture medium inoculated with *B. thetaiotaomicron* as the sole continuous source of hydrogen and carbon dioxide [16]. Subsequently, Guindo et al. improved the double chamber culture technique by proposing a new concept, based on the production of hydrogen through a chemical reaction between acetic acid and iron filings [91].

In the oral cavity, methanogenic archaea are part of the commensal microbiota, especially in individuals who smoke [92]. They are also associated with periodontal pathologies and especially with periodontitis [93] and peri-implantitis [94]. However, they have also been detected in inflammatory and infected dental pulp [95]. *Methanobrevibacter oralis*, *M. smithii* and *Methanobrevibacter massiliense* are the sole methanogenic archaea cultured from oral cavity specimens, mainly saliva and dental plaque (Figure 1) [93]. Furthermore, the discovery of *Nanopusillus massiliensis* in dental plaque samples and its co-culture with *M. oralis* revealed the *Nanoarchaeota* phylum in the human oral cavity [96]. Finally, other archaea species have only been detected by molecular tools, including *Methanobacterium congolense*, *Methanocellus bourgensis*, Candidatus *Nitrosphaera everglandensis*, *Methanosarcinia mazei* [88,97] as well as non-methanogenic species belonging to the phylum *Thermoplasmata* [98,99].

Despite recent investigations regarding the culture of archaea, as well as technological advances and new culture methods, there remains a significant discord between the diversity of archaea in the gut microbiota and that of the oral cavity. To date, only three archaea species have been isolated from the oral cavity compared to seven from the digestive tract. This difference may be explained by the interest accorded to the gut microbiota recent years, particularly in the context of the Human Microbiome Project, which has focused on characterisation of the human gut microbiota [100]. In contrast, the oral microbiota remains neglected by microbiologists, and a lack of investigation and research work is notable, especially in the field of the culture of methanogenic archaea [83,88]. Additional efforts are needed to extend the repertoire of these microorganisms by first using molecular detection and diagnostic tools and then culture-based methods available in the literature to isolate them in order to better understand their metabolism and their involvement in human health and pathologies.

### 2.7. Fungi and Yeasts of the Oral Cavity through the Example of Candida

Fungi are a minor component of the oral microbiota. The diversity of this component is mainly represented by three genera, namely *Candida*, *Aspergillus*, and *Cryptococcus* (Figure 1) [101]. These commensal microorganisms are regularly present in low concentrations without causing infection, but they can become opportunistic pathogens when their host is immunodeficient [102]. The majority of oral fungal infections are forms of candidiasis [103]. These infections may be superficial and affect the mucosa, or they may penetrate the epithelium and be disseminated through the haematogenous pathway with serious consequences. Mucosal infections are seen in neonates and the elderly, two groups with suboptimal immune function [101]. They also affect people with suppressed immune systems [103].

Since the discovery of the first microbes [104], many techniques have been developed to study and cultivate fungi and yeasts from the oral cavity. Indeed, multiple references are proposed by manufacturers to enable an easy and selective culture of these microorganisms, including Sabouraud dextrose agar, malt agar, potato dextrose agar, CZAPEK, Colombia agar, Dixon agar, modified Leeming Notman agar, YPD medium and glycine-vancomycin-polymyxin B agar [32].

Recently, a new culture medium called FastFung^®^ was developed by the Culturomics, Fungi and Yeasts Team at the IHU Méditerranée Infection, Marseille, with the objective of cultivating fastidious fungi [105]. The FastFung^®^ medium was compared to the standard Sabouraud agar medium for the culture of 98 fungal and 20 bacterial strains. The positive culture rate of fungal strains was 100% compared to 95% for Sabouraud, and the inhibition of bacterial strains was 100% compared to 20% for Sabouraud medium [105].

The FastFung^®^ culture medium was subsequently tested for the specific culture of *Malassezia* yeasts [106]. The aim of this study was to compare the performance of three skin sampling methods and two culture media for the detection of *Malassezia* yeasts by culture from human skin samples. The FastFung^®^ medium showed promising results for the specific culture of *Malassezia* yeast and was proposed for direct application in routine laboratory clinical diagnosis [106].

The FastFung^®^ medium is a promising culture medium due to the results provided for the culture of fungi and yeasts from clinical samples. Therefore, this medium can be adapted for future studies for the cultivation of fungi and yeasts from the oral microbiota.

### 2.8. Protozoa of the Oral Cavity: Through Amoebae and Trichomonas

Protozoa are microscopic, unicellular eukaryotes. Each protozoan typically exists as an independent cell, living freely as a phagotrophic microorganism [107]. In some species, the cells join together to form colonies. Many protozoa are pathogenic and can cause diseases such as malaria, while others are commensals in the digestive tract of ruminants and wood-eating insects [107].

Protozoa are also part of the oral microbiota, of which *Entamoeba gingivalis* is the most frequent representative, followed by *Trichomonas tenax* (Figure 1) [108]. Oral protozoan infections have been shown to be mainly associated with periodontal disease [8]. Indeed, their prevalence is much higher in patients with periodontal disease than in those with healthy periodontium [108]. Currently, the detection of oral cavity protozoa is mainly performed by direct microscopy or by PCR targeting the 18S gene [8,109]. Culture-based approaches applied to these organisms are very rare in the recent literature. The few studies which have been published use a minimal culture medium with an agar base comprised between 1.5% and 2% [110]. The ATCC 1171 TYGM-9 culture medium (https://www.atcc.org/products/30956; accessed on 23 January 2023) adopted from the work of Clark et al. in 1992 and 1997, remains the reference medium for the culture of protozoa from the human oral cavity [111,112,113]. New investigations are, therefore, necessary to improve the culture of these fastidious organisms and new culture media must be optimised to broaden the research spectrum and isolate new species that have remained uncultured.

**Table 1 microorganisms-11-00836-t001:** Update and description of culture techniques and their research perspectives.

Techniques	Usual Denomination	Applications and Perspectives	References
Next-generation sequencing	NGS	Molecular-based study of the microbial diversity of diverse ecological niches.	[24,25,26,27]
Culturomics	Culturomics	Culture-based study of the microbial diversity of the human microbiota.	[18,19,20,30]
Rapid culture of anaerobic bacteria using YCFA medium	Rapid culture	Culture-based study of stool samples.	[36]
Blood culture bottles supplemented with Rhumen	Rapid culture	Culture-based study of stool samples.	[36]
Alcohol decontamination	NO	Spore-forming bacteria.	[35,42]
Thermal shock	NO	Spore-forming bacteria.	[18]
Antioxidants	NO	Improvement of anaerobic conditions for aero-intolerant bacteria culture.	[16,45,46]
Co-culture	NO	Improvement of fastidious bacteria culture.	[16,49,50,51]
Selective media	Antibiotics and phages	Selective culture of antibiotics and phase-resistant organisms.	[20,54,56]
The genomic-reverse culture technique	The genomic-reverse	Saccharibacteria TM7 group.	[57,58]
Oral Treponema enrichment	T-Raoult medium	*Treponema* spp.	[71]
Specific media for Treponema culture	OMIZ-WP = OMIZ-Pat	*Treponema* spp.	[79]
	OTEB	*Treponema* spp.	[78]
	NOS	*T. pectinovorum* *T. denticola* *T. socranskii* subsp. *Socranskii* *T. vincentii*	[77]
	OTI	*T. pectinovorum*	[76]
	Oral spirochete medium	*T. denticola*	[114]
	PDDTp	*T. phagedenis* *T. denticola*	[115]
	Spirolate Medium	*T. vincentii*	[116]
	Thioglycolate medium	*Treponema reiter* *T. phagedenis* *T. refringens* *T. vincentii*	[117]
	Pre-reduced *T. denticola* broth	*T. denticola*	[118]
	Growth Medium (GM-1)	*T. denticola*	[74]
	Medium 10 (M10)	*T. socranskii*	[75]
	Human oral medium (HO)	*T. phagedenis* *T. refringens* *T. denticola* *T. vincentii*	[119]
Hungate roll tubes	Roll tubes	Methanogenic archaea culture. and strictly anaerobic bacteria.	[44]
Double chamber technique	SAB-medium	Methanogenic archaea culture.	[16]
Modified double chamber technique		Methanogenic archaea culture.	[97]
FastFung^®^	FastFung^®^	Fungi and yeasts.	[105]
The ATCC 1171 TYGM-9 culture medium	ATCC 1171	Protozoa of the oral cavity.	[110]

NO: unspecified.

## 3. Discussion

Since the discovery of the first microbes by Anthonie van Leeuwenhoek in the 17th century (1683) [104], microbial culture was one of the first methods used to study the human microbiota [18]. Since then, microbial culture has greatly contributed to improving the scientific and medical knowledge of the microbes studied as well as of the diseases they provoke [19]. Microbial culture has long been considered tedious, costly and time consuming [20,21]. As a result, microbiologists have gradually abandoned it in favour of new molecular approaches such as PCR techniques and metagenomics [20,21]. Indeed, metagenomics has been considered as one of the most promising techniques for studying microorganisms without culturing them. It was first applied in 2005 to study the human intestinal microbiota with the prospect of revealing all remaining uncultured microbes [18]. Subsequently, culture-based studies were gradually abandoned and metagenomics became the reference method used by microbiologists to study complex ecosystems [24]. However, this technique presented many biases related to the extraction methods and the type of primers used for the amplification of genomic DNA, which made the results obtained non-reproducible [120]. In addition, a depth bias makes all species with a population below 1 × 10^5^ or even 1 × 10^6^ per gram of sample undetectable by this technique [19,24,120]. Nevertheless, metagenomics has contributed to the expansion of knowledge about the uncultured part of the human microbiota, which does not correspond to any known microbe, commonly referred to as “dark matter” [19,20]. It is a question of an uncountable number of sequences not assigned to any known microorganism, which metagenomics is unable to identify [19,20].

In recent years, microbial culture has experienced a re-emergence thanks to environmental microbiologists who have developed culture conditions close to the original environment of bacteria in order to facilitate their cultivation [18]. Numerous approaches have subsequently proven their value in microbial culture, such as microbial culturomics, which has made considerable progress in the field of culture and especially in the repertoire of human microbiota, enabling the culture of more than 2770 spaces of bacteria, archaea, fungi and yeasts, as well as unicellular eukaryotes [31]. In contrast, few studies have focused on the oral microbiota by culture, and studies performed in this field remain limited. Indeed, the oral cavity hosts a complex microbiota composed of more than 700 different bacterial species colonising the surface of the teeth and the oral mucosa [22]. After the gut microbiota, it is the second microbiota in humans in terms of number and diversity [22].

The study of the human microbiota has become a major public health issue given its involvement in human health and disease, which goes beyond our understanding. Several studies in the literature have clearly demonstrated its involvement not only in disease but also in human health. Its involvement in the regulation of the response to anti-cancer treatments has been demonstrated [121,122,123]; as well as the participation in the metabolic homeostasis of certain species like *Akkermansia muciniphila*, which could have a protective effect against metabolic disorders [124]. Indeed, the involvement of *Faecalibacterium prausnitzii* and *A. muciniphila*, two of the most abundant bacteria in the human gastrointestinal tract, in the marked attenuation of colitis, weight loss, and decrease of pro-inflammatory cytokines and increase of anti-inflammatory cytokines in patients with chronic inflammatory bowel disease has been demonstrated [125,126]. In the oral cavity, numerous studies have described a protective role attributed to the oral microbiota. According to Sekirov et al., 2010, the microbiota may provide its host with a physical protective barrier against incoming pathogens by competitive exclusion [7]. Indeed, many actions of the microbiota can have a beneficial effect on its host, such as the occupation of attachment sites, the consumption of nutrient sources and the production of antimicrobial substances [7]. The microbiota also acts on the immune system of its host by stimulating the production of various antimicrobial compounds such as defensins, cathelicidins and C-type lectins, known to be produced generally in the digestive tract of mammals [7,127].

The oral microbiota has also been incriminated in certain infectious and tumoral pathologies of the oral cavity as well as in numerous systemic pathologies such as diabetes, cardiovascular diseases and certain distant tumour illnesses [23].

It now seems clear that the oral microbiota plays an essential role in the maintenance of oral health in particular and of human health in general. The alteration of this microbiota has often been associated with oral and systemic pathologies. However, the role of the oral microbiota in the onset and development of these diseases remains unresolved, and further investigations are needed to characterise the oral microbiota and elucidate its involvement in human and oral pathologies. Culture-based studies offer good potential in this regard, and microbial culture still has much to contribute to the study of microbiota in order to obtain clinical isolates, to study their metabolism and their behaviour towards antimicrobial agents used in clinical practice, as well as to sequence their genome to better understand their implication in the process of oral infectious diseases.

## 4. Patents

S.K. is co-inventor of a patent on the culture of anaerobic bacteria (CAS 28-FR1757574); S.K. is co-inventor of a patent for the preservation of bacteria (1H53 316 CAS 25). S.K. is co-inventor of patent: “Milieu et procede de culture et extraction des archaea methanogenes” (FR1254779A). S.K. is co-inventor of a patent on the use of antioxidants to aerobically cultivate anaerobic bacteria and methanogenic archaea (1H52437 cas 32FR). S.K. and S.B. are co-inventors of a patent on the use of vinblastine or vincristine to improve the culture of treponemes (FR 19 13945).

## Figures and Tables

**Figure 1 microorganisms-11-00836-f001:**
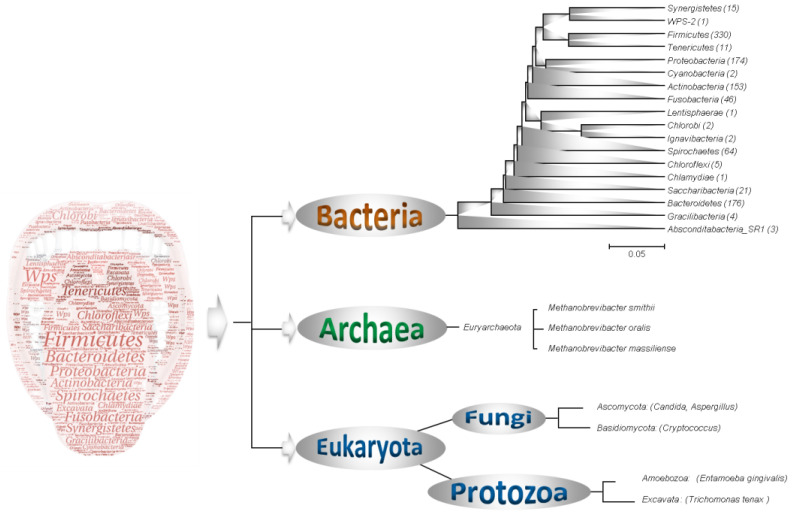
Phylogenetic diversity of the 19 phyla isolated or detected at least once in the human oral cavity. The number of species per phylum is presented between parents in the phylogenetic tree. The illustration on the left of the figure was made using the online tool wordle (www.wordle.net; accessed on 23 January 2023); the size of the name of each phylum is proportional to the number of species it contains. The phylogenetic tree was created using MEGA software. Figure 1 was built according to data collected from the Human Oral Microbiome Database; https://www.homd.org/; accessed on 23 January 2023.

**Figure 2 microorganisms-11-00836-f002:**
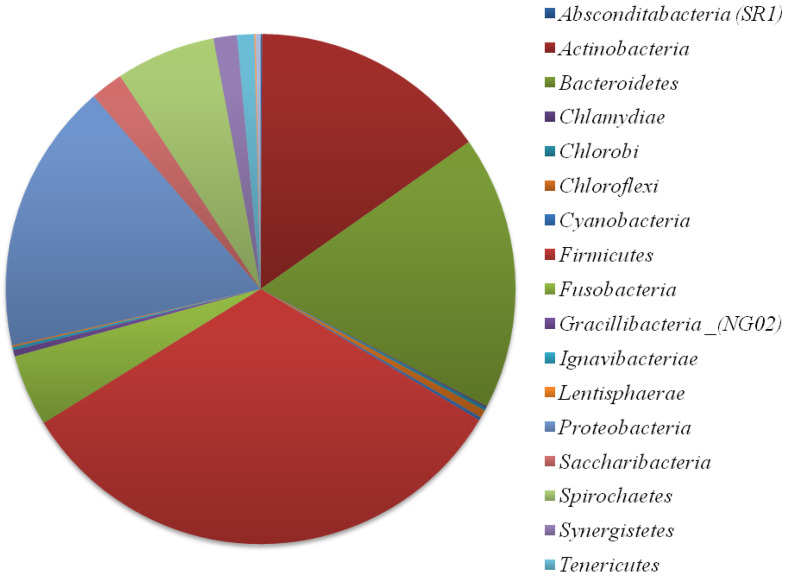
Quantitative distribution of cultured and uncultured bacterial diversity composing the oral microbiota represented by number of species composing each phylum. Each phylum is represented by a different colour, as shown on the right. (Figure 2 was built according to data collected from the Human Oral Microbiome Database; https://www.homd.org/; accessed on 23 January 2023).

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
