# Peer review of "Culturing the Human Oral Microbiota, Updating Methodologies and Cultivation Techniques"

_microorganisms, 2023, doi:10.3390/microorganisms11040836_

Round 1
Reviewer 1 Report (Previous Reviewer 1)
I would like to appreciate the authors' effort to rewrite parts of the manuscript that other reviewers recommend changing, correcting, or adding the missing information.
Author Response
Reviewer 1:
I would like to appreciate the authors' effort to rewrite parts of the manuscript that other reviewers recommend changing, correcting, or adding the missing information.
- We thank the reviewer for his positive comments.
Reviewer 2 Report (Previous Reviewer 2)
In this revised version, the authors did not sufficiently address most of the issues raised by this reviewer about the original version. The proposal of this manuscript is to address the application of cultivation techniques to improve the knowledge about the composition of the oral microbiota. However, the authors could not provide a sufficient description of major traits of the oral microbiomes associated with homeostasis and disease. There is a lack in cohesion of the major topics addressed, and most of the specific issues raised about the original article were only addressed by removing parts of the text. Issues related to poor explanation of data represented in the Figures also remain.
Author Response
Reviewer 2:
In this revised version, the authors did not sufficiently address most of the issues raised by this reviewer about the original version. The proposal of this manuscript is to address the application of cultivation techniques to improve the knowledge about the composition of the oral microbiota. However, the authors could not provide a sufficient description of major traits of the oral microbiomes associated with homeostasis and disease. There is a lack in cohesion of the major topics addressed, and most of the specific issues raised about the original article were only addressed by removing parts of the text. Issues related to poor explanation of data represented in the Figures also remain.
- We thanks the reviewer for their comments, we are preparing a future review to address this topic in greater detail, it will be submitted soon and we will be honoured if the reviewer will accept to review it.
Reviewer 3 Report (New Reviewer)
This is an interesting manuscript, which is of great interest of everybody working on commensal and oral pathogenic oral bacteria and on treatment of derived diseases. The reviewer find the manuscript adequate and very interesting, and it is that it shows the relevance of the cultivation technique, and that it has certainly lost in the face of new independent cultivation techniques such as massive sequencing. However, as indicated in the work, all techniques have their limitations, both cultivation techniques and those independent of it. As authors well indicate, when working with microorganisms, especially at a clinical way, we realize the need to combine molecular techniques such as NGS with the culture technique, to overcome the limitations of these techniques and have a global and complete vision of the composition of the microbiota of a given ecological niche and, on many occasions, characterize which or which of these microorganisms are causing damage to the health of the individual.
General comments
As general comments I would like to pose a question to the authors and make an indication of an important error detected. First, the question is that throughout the text they talk about metagenomics techniques, but referring to techniques used to characterize bacteria in an ecological niche, wouldn't it be more appropriate to talk about metataxonomics instead of metagenomics?
Regarding the detected error: There is an important error in the manuscript that must be corrected. The bibliographical references should be carefully reviewed, since the first paragraph of the discussion refers to citations 121, 122, 123 and 124, but these references do not exist in the references section.
Specific Comments
Pag 1, lines 17-19: … “We discuss the use of culture media to culture fastidious bacteria, methods which opened the door to the use of culture as a diagnostic tool in hospital laboratories”.
The culture technique is a reference technique used routinely in hospitals, but in hospitals, often for urgent treatments against a certain organism, rapid tests such as PCR can save lives. In my opinion, I would withdraw this reference to use in hospitals, since later in the manuscript you do not make any explicit reference to use in said centers.
Pag 2, lines 46-47: … “on social, economic, dietary and nutritional factors, as well as sedentary lifestyle, autonomy, hygiene compliance, smoking, and alcohol consumption” ….
The genetic factor and physiological factors should be included, such as the hormonal changes associated with pregnancy.
Pag 2, line 52: the term “flora" is no longer used, please make a better reference to “microbiota” throughout the document.
Pag 4, line 145: “YCFA (Yeast extract-Casein hydrolysate Fatty Acids)” … In general, the developed name must be indicated first, and then, in parentheses, its acronym.
Pag 7: Section 2.5 describes spirochetes in detail, but does not inform the reader about culture media. You indicate that there are many tested, but you should indicate which one or which you think are the best, having done the literature review. Nor is it very clear to the reader in table 1 to which they refer.
Author Response
Reviewer 3
This is an interesting manuscript, which is of great interest of everybody working on commensal and oral pathogenic oral bacteria and on treatment of derived diseases. The reviewer find the manuscript adequate and very interesting, and it is that it shows the relevance of the cultivation technique, and that it has certainly lost in the face of new independent cultivation techniques such as massive sequencing. However, as indicated in the work, all techniques have their limitations, both cultivation techniques and those independent of it. As authors well indicate, when working with microorganisms, especially at a clinical way, we realize the need to combine molecular techniques such as NGS with the culture technique, to overcome the limitations of these techniques and have a global and complete vision of the composition of the microbiota of a given ecological niche and, on many occasions, characterize which or which of these microorganisms are causing damage to the health of the individual.
General comments
As general comments I would like to pose a question to the authors and make an indication of an important error detected. First, the question is that throughout the text they talk about metagenomics techniques, but referring to techniques used to characterize bacteria in an ecological niche, wouldn't it be more appropriate to talk about metataxonomics instead of metagenomics?
- We thank the reviewer for this comment, the references cited in this paper indeed refer to techniques used to characterize bacteria in an ecological niche, but also to methagenomics techniques.
Regarding the detected error: There is an important error in the manuscript that must be corrected. The bibliographical references should be carefully reviewed, since the first paragraph of the discussion refers to citations 121, 122, 123 and 124, but these references do not exist in the references section.
- The reviewer is right, we apologize for this error, this part is not part of the current version of the paper, and it’s now removed from this new version.
Specific Comments
Pag 1, lines 17-19: … “We discuss the use of culture media to culture fastidious bacteria, methods which opened the door to the use of culture as a diagnostic tool in hospital laboratories”.
The culture technique is a reference technique used routinely in hospitals, but in hospitals, often for urgent treatments against a certain organism, rapid tests such as PCR can save lives. In my opinion, I would withdraw this reference to use in hospitals, since later in the manuscript you do not make any explicit reference to use in said centers.
- The reviewer is right; we have now removed this reference from the current version of the manuscript
Pag 2, lines 46-47: … “on social, economic, dietary and nutritional factors, as well as sedentary lifestyle, autonomy, hygiene compliance, smoking, and alcohol consumption” ….
The genetic factor and physiological factors should be included, such as the hormonal changes associated with pregnancy.
- We thanks the reviewer for this interesting note missed in this manuscript, genetic factor and physiological factors such as the hormonal changes associated with pregnancy were now included Line 47 as well as appropriate references [9] and [11].
Pag 2, line 52: the term “flora" is no longer used, please make a better reference to “microbiota” throughout the document.
- The reviewer is right; we now refer to the microbiota in the whole manuscript.
Pag 4, line 145: “YCFA (Yeast extract-Casein hydrolysate Fatty Acids)” … In general, the developed name must be indicated first, and then, in parentheses, its acronym.
- The reviewer is right, the developed name is now indicated first, and then, in parentheses, its acronym Line 147.
Pag 7: Section 2.5 describes spirochetes in detail, but does not inform the reader about culture media. You indicate that there are many tested, but you should indicate which one or which you think are the best, having done the literature review. Nor is it very clear to the reader in table 1 to which they refer.
- The reviewer is right, we have added now 10 lines in the section “Treponemes of the oral cavity” Lines 334-342 to describe Treponema culture media and they are now listed in Table 1.
Round 2
Reviewer 3 Report (New Reviewer)
Thank you very much for taking into consideration the clarifications that I indicated.
the manuscript is adequate, only you should review line 340, page 9, which begins with a punctuation mark.
Author Response
We Thanks the reviewer for this comment, the error line 340, page 9 is now corrected
This manuscript is a resubmission of an earlier submission. The following is a list of the peer review reports and author responses from that submission.
Round 1
Reviewer 1 Report
Dear authors,
After a precise evaluation of the manuscript, I consider the presented review as well-written and structured. The authors described step-by-step methodologies used for the detection of oral microbes. I observed only minor grammar errors, mainly in the names of microorganisms (missing italics or spelling for example line 120: Porphyromonas ingivalis). I recommend a double check of it.
Reviewer 2 Report
Review of MDPI - Microorganisms-2009378 entitled “Culturing the human oral microbiota, updating methodologies and cultivation techniques“, by Saber Khelaifia and Gerard Aboudharam as corresponding authors.
General Comments.
This review article addresses current methods of culturing microorganisms that could be applied for expanding the knowledge on the microbial composition of the human oral microbiome, as well as its functions in health and disease. The text looks straightforward and the manuscript contribute to the field of oral microbiology, by highlighting novel culturing techniques developed for isolation of microorganisms from complex samples. However, this manuscript unfortunately lacks rigor in acknowledging studies of the field and in supporting statements. Misinterpretation and/or mistakes in the selection of the acknowledged studies were also detected. The authors also oversimplify the pathogenesis of dental caries and adult chronic periodontitis. Some major points are indicated in the specific comments. Improvements in writing are also required.
Specific comments:
1) The Abstract needs to be carefully revised to reflect the major points addressed in this review. There are sentences poorly constructed, which are elusive or do not specifically address the major topics of the manuscript. For example, the sentences of l. 15-17 (“Despite recent investigations… gut microbiota and that in the oral cavity“) need to be removed or modified.
2) The authors need to acknowledge the complexities of the oral cavity. There are numerous site-specific niches in the oral environment, which complicates the study and definition of the “oral microbiome”. For example, solely for dental sites, there different sub-sites which remarkably differ in microbial composition: sub-gingival versus supra-gingival surfaces; smooth surfaces (lingual, buccal/labial versus proximal) versus occlusal surfaces. Moreover, studies on the oral microbiome are not really “rare”, although several may be limited to a restricted number of oral niches or to specific oral conditions, age groups, ethnic groups, etc. A large number of studies on the oral microbiome are missing in this review. Please, look at some few examples:
Becker MR, Paster BJ, Leys EJ, Moeschberger ML, Kenyon SG, Galvin JL, Boches SK, Dewhirst FE, Griffen AL. Molecular analysis of bacterial species associated with childhood caries. J Clin Microbiol. 2002 Mar;40(3):1001-9. doi: 10.1128/JCM.40.3.1001-1009.2002. PMID: 11880430; PMCID: PMC120252.
Kazor CE, Mitchell PM, Lee AM, Stokes LN, Loesche WJ, Dewhirst FE, Paster BJ. Diversity of bacterial populations on the tongue dorsa of patients with halitosis and healthy patients. J Clin Microbiol. 2003 Feb;41(2):558-63. doi: 10.1128/JCM.41.2.558-563.2003. PMID: 12574246; PMCID: PMC149706.
Aas JA, Griffen AL, Dardis SR, Lee AM, Olsen I, Dewhirst FE, Leys EJ, Paster BJ. Bacteria of dental caries in primary and permanent teeth in children and young adults. J Clin Microbiol. 2008 Apr;46(4):1407-17. doi: 10.1128/JCM.01410-07. Epub 2008 Jan 23. PMID: 18216213; PMCID: PMC2292933.
Zaura E, Keijser BJ, Huse SM, Crielaard W. Defining the healthy "core microbiome" of oral microbial communities. BMC Microbiol. 2009 Dec 15;9:259. doi: 10.1186/1471-2180-9-259. PMID: 20003481; PMCID: PMC2805672.
Colombo AP, Boches SK, Cotton SL, Goodson JM, Kent R, Haffajee AD, Socransky SS, Hasturk H, Van Dyke TE, Dewhirst F, Paster BJ. Comparisons of subgingival microbial profiles of refractory periodontitis, severe periodontitis, and periodontal health using the human oral microbe identification microarray. J Periodontol. 2009 Sep;80(9):1421-32. doi: 10.1902/jop.2009.090185. PMID: 19722792; PMCID: PMC3627366.
Dewhirst FE, Chen T, Izard J, Paster BJ, Tanner AC, Yu WH, Lakshmanan A, Wade WG. The human oral microbiome. J Bacteriol. 2010 Oct;192(19):5002-17. doi: 10.1128/JB.00542-10. Epub 2010 Jul 23. PMID: 20656903; PMCID: PMC2944498.
Tanner AC, Mathney JM, Kent RL, Chalmers NI, Hughes CV, Loo CY, Pradhan N, Kanasi E, Hwang J, Dahlan MA, Papadopolou E, Dewhirst FE. Cultivable anaerobic microbiota of severe early childhood caries. J Clin Microbiol. 2011 Apr;49(4):1464-74. doi: 10.1128/JCM.02427-10. Epub 2011 Feb 2. PMID: 21289150; PMCID: PMC3122858.
Yost S, Duran-Pinedo AE, Teles R, Krishnan K, Frias-Lopez J. Functional signatures of oral dysbiosis during periodontitis progression revealed by microbial metatranscriptome analysis. Genome Med. 2015 Apr 27;7(1):27. doi: 10.1186/s13073-015-0153-3. Erratum in: Genome Med. 2015;7(1):111.
Mark Welch JL, Rossetti BJ, Rieken CW, Dewhirst FE, Borisy GG. Biogeography of a human oral microbiome at the micron scale. Proc Natl Acad Sci U S A. 2016 Feb 9;113(6):E791-800. doi: 10.1073/pnas.1522149113. Epub 2016 Jan 25. PMID: 26811460; PMCID: PMC4760785.
Duran-Pinedo A, Solbiati J, Teles F, Teles R, Zang Y, Frias-Lopez J. Long-term dynamics of the human oral microbiome during clinical disease progression. BMC Biol. 2021 Nov 6;19(1):240. doi: 10.1186/s12915-021-01169-z. PMID: 34742306; PMCID: PMC8572441.
Fellows Yates JA, Velsko IM, Aron F, Posth C, Hofman CA, Austin RM, Parker CE, Mann AE, Nägele K, Arthur KW, Arthur JW, Bauer CC, Crevecoeur I, Cupillard C, Curtis MC, Dalén L, Díaz-Zorita Bonilla M, Díez Fernández-Lomana JC, Drucker DG, Escribano Escrivá E, Francken M, Gibbon VE, González Morales MR, Grande Mateu A, Harvati K, Henry AG, Humphrey L, Menéndez M, Mihailović D, Peresani M, Rodríguez Moroder S, Roksandic M, Rougier H, Sázelová S, Stock JT, Straus LG, Svoboda J, Teßmann B, Walker MJ, Power RC, Lewis CM, Sankaranarayanan K, Guschanski K, Wrangham RW, Dewhirst FE, Salazar-García DC, Krause J, Herbig A, Warinner C. The evolution and changing ecology of the African hominid oral microbiome. Proc Natl Acad Sci U S A. 2021 May 18;118(20):e2021655118. doi: 10.1073/pnas.2021655118. PMID: 33972424; PMCID: PMC8157933.
3) There is a number of statements, which are not supported by scientific studies. For example: l. 125-122 “Thus, new concepts have emerged …”; l. 553-557 “Individuals who have acidophilic bacteria in their microbiota,…”
4) L. 140-141; Figure 3. It is not possible to identify source of data used to build Figure 3. The legend of this Figure is not elucidative. The same for Figure 2.
5) L. 194-198. The authors need to provide a more complete description of “Culturomics”.
6) L. 221 to 225. Please, provide an estimated number of species (or operational taxonomic units) of the gut microbiome for more accurate comparisons.
7) L. 251-252. Please, provide a rationale supporting the suggestion that YCFA medium would be useful for exploring the oral microbiome. Perhaps defining major genera/species identified with this approach in the gut. This information could be shown in a Figure.
8) L. 259-262. It seems that there is a mistake here. At least in study # 53, non-spore forming Gram-positive bacteria were reported (not spore-forming).
9) L. 299-300. This sentence looks vague and exaggerated. Please, emphasize the power of this method, by informing the number of species/isolates identified and advantages of this approach.
10) L. 370-371. It would be better to explain the study of ref. 73 to replace this statement.
11) l. 456-458. Reference #103 is not appropriate for supporting this sentence. However, several studies addressing archae in oral sites are acknowledged in this study; these should be acknowledged. Please, also look at the following review article for updating manuscript information:
de Cena JA, Silvestre-Barbosa Y, Belmok A, Stefani CM, Kyaw CM, Damé-Teixeira N. Meta-analyses on the Periodontal Archaeome. Adv Exp Med Biol. 2022;1373:69-93. doi: 10.1007/978-3-030-96881-6_4. PMID: 35612793.]
12) L. 553-557. The authors oversimplify the pathogenesis of dental caries and do not provide references supporting their statements. They further seems to ignore the importance of the extracellular matrix in biofilm cariogenicity, as well as the major microbial and environmental pressures shaping its structure and composition. Figure 1 does not depict the central role of the extracellular matrix in biofilm cariogenic potential, as well.
Minor points:
- Abstract: should be “whereas studies” instead of “and work”.
- Abstract: should be “still limited” instead of “very rare”.
- Table 1. 2nd column does not refer to "abbreviations" only. Perhaps, use "Usual denomination" instead of "abbreviations". Also the first technique indicated in this table is not a culture-base, so could be removed. Finally, "Selective media" should be “Media supplemented with antibiotics and/or phages”.
- L. 520. It should read "subjected" instead of "subject".
- L. 533. "Microbial succession" instead to "Attachment and colonisation" would be more appropriated.
- Author Contributions. This section needs to be carefully revised. There is no co-author with the initials X.X. .
Reviewer 3 Report
This manuscript intended to give an overview of the culture methods available to study oral bacteria.
However, clear statement of this goal is not properly done at the beginning of the paper, which leaves the reader confused throughout the reading. Only at lines 203 -205 do we see such an attempt.
For example, the inclusion of sections on caries and periodontal disease are not justified in my opinion. Furthermore, the section on the microbial causes of periodontal disease is not up to standard. There are key concepts missing such as the ecological plaque hypothesis and the keystone species concept.
I have also found that the references are sometimes outdated (9 and15 for example) and not appropriate in other instances.
Figures have some errors: Figure 1 I dont agree with the term carbohydrates as a cause of the disequilibrium in the supragingival microbiota. It should be fermentable sugars. Figure 2 there are at least 2 spelling mistakes noted in the file attached. Figure 3 does not have a clear legend.
Section 3, which should be the actual manuscript, needs to be better organized. It is no clear to me the rationale for having six subsections with such diverse content. Maybe dividing it in techniques on the one hand and specific fastidious groups on the other (those requiring very specific approaches).
Table 1 is well structured and can be the base of the manuscript.
In the discussion more information is presented, and the relevance is not discussed (first paragraph of the discussion).
I suggest that authors re Write this manuscript focusing on the culture techniques themselves and then, in the discussion reflect on how these techniques improve our knowledge of the oral ecosystem.
